# Recovery of Routine Immunisation: Mapping External Financing Opportunities for Reaching Zero-Dose Children

**DOI:** 10.3390/vaccines11071159

**Published:** 2023-06-26

**Authors:** Sarah Tougher, Nikhil Mandalia, Ulla Kou Griffiths

**Affiliations:** 1Independent Researcher, Winnipeg, MB R3N 1P6, Canada; 2United Nations Children’s Fund (UNICEF) Programme Group, New York, NY 10017, USA

**Keywords:** vaccine, immunisation, financing, development assistance for health, universal coverage, zero dose

## Abstract

The COVID-19 pandemic has precipitated large declines in childhood vaccination coverage and, consequently, substantial increases in the number of zero-dose children. To effectively respond to these declines, it is necessary to direct resources for recovery. We mapped active external financing for immunisation and primary healthcare in 20 countries with the highest numbers of zero-dose children to promote transparency and donor coordination. We found that countries have disparate access to external financing, with the two upper-middle-income countries (Brazil and Mexico) only having access to loans from the International Bank for Reconstruction and Development. Domestic resource mobilization is, therefore, key in these two countries, although fiscal space is likely constrained. Four additional countries (Angola, Indonesia, Philippines, and Vietnam) do not have allocations from Gavi, the Vaccine Alliance for Health Systems Strengthening, or Equity Accelerator Funding, but are eligible for support under Gavi’s Middle-Income Countries Approach. Our methods, which focus on current donor financing, are novel and reveal substantial variations in access to external financing to support immunisation in high-burden countries. The available data differ considerably across financing mechanisms, making it difficult to synthesise the results across funding sources.

## 1. Introduction

In July 2022, WHO and UNICEF published updated vaccination coverage estimates, which revealed the largest sustained decline in childhood vaccination coverage in 30 years [1]. It is estimated that 25 million children missed out on one or more doses of diphtheria–tetanus–pertussis (DTP) in 2021; of these, 18 million children did not receive even one dose of the DTP vaccine. This population group, known as zero-dose children, represents those who did not receive a single vaccine.

In response to this, national governments need to revise their immunisation strategies to ensure the development and implementation of interventions that can reach zero-dose children and raise vaccination coverage in a pro-equity manner. UNICEF and WHO support the development of an immunisation recovery plan covering 20 countries with the highest numbers of zero-dose children. As part of the planning process, there is a need to understand what financial resources may be available from external sources, and the extent to which these can be aligned with the zero-dose agendas of the countries.

The overall objective of the present finance mapping exercise was to strengthen the transparency of external funding supporting immunisation activities in countries with high numbers of zero-dose children. The intention was to ensure that countries are empowered with information on available donor funding, which can facilitate efforts to coordinate and align resources with revised national priorities that include interventions targeting zero-dose children.

## 2. Methods

For the 20 countries with the largest numbers of zero-dose children in 2021 (Afghanistan, Angola, Brazil, Cameroon, Chad, the Democratic People’s Republic of Korea, Democratic Republic of Congo, Ethiopia, India, Indonesia, Madagascar, Mexico, Mozambique, Myanmar, Nigeria, Pakistan, the Philippines, Somalia, Tanzania, and Vietnam), profiles were developed detailing external resources allocated to each country to support immunisation or health system strengthening activities. These country profiles give a high-level understanding of the immunisation and health financing landscapes and provide an entry point for governments to engage with donor organizations on their programmatic priorities.

### Data Collection and Analysis Approach

A ‘top-down’ approach was followed, through which external funding sources/organizations were surveyed and asked to provide relevant information on funding channeled towards each country. The engagement occurred through the central/headquarters of each organization, leveraging centralized data records on the funding provided to each country. This approach was determined to be less resource- and time-intensive compared to a ‘bottom-up’ approach, which would entail engagement with organizations via their country offices and in-country implementing partners.

Respondents from key organisations that finance immunisation and healthcare were selected based on UNICEF’s experience in tracking financing for COVID-19 vaccine delivery for the COVID-19 Vaccine Financial Monitoring (C19VFM) database [2].

Respondents were requested to report active grant funding for routine immunisation, vaccination campaign activities, COVID-19 vaccine delivery, and immunisation and healthcare system strengthening efforts. The scope of the data requested included all funding commitments and disbursements, as well as a list of in-country implementing partners. In practice, many financing sources were not able to report all aspects requested, including disbursements of active financing, due to limitations in the data readily available within internal grant management systems.

For the World Bank, a list of active health-related projects that potentially support immunisation-related activities was provided by the bank staff from a recent portfolio review of immunisation-related investments [3]. The total number of health-related projects, including COVID-19-related health projects, and total commitments across those projects were reported. COVID-19-related projects were reported separately because unused funds could potentially be repurposed to support the recovery and strengthening of routine immunisation. However, this would need to be assessed on a project-by-project basis and would depend on country priorities and other constraints.

Gavi-supported countries have allocations through several financing streams, including Health Systems Strengthening (HSS), Equity Accelerator Funding (EAF), COVID-19 Vaccine Delivery Support (CDS), Cold Chain Equipment Optimisation Platform (CCEOP), and Targeted Country Assistance (TCA). Five-year allocations (2021–2025) for the Gavi 5.0 period for each of the financing streams were calculated based on allocation formulas. Since HSS and EAF, as well as up to 50% of funds from the third window of CDS support (CDS3), could be used to recover and strengthen routine immunisation, potentially unallocated Gavi funds through HSS, EAF, and 50% of CDS3 funds per country were calculated by subtracting approved amounts for 2021–2023 from the allocation caps and ceilings. These are ‘potentially unallocated’ funds, as countries may have already submitted applications that are in the process of being reviewed, but not yet approved at the time of data collection, and the countries will most likely ultimately use these funds. To facilitate comparisons across countries, potentially unallocated Gavi funds were standardized per zero-dose child in 2021.

## 3. Results

### 3.1. Country Characteristics

The basic characteristics of the 20 countries with the highest numbers of zero-dose children are presented in Table 1. These countries have disparate contexts in terms of their economic circumstances, level of health expenditures, and vaccination coverage. The 20 countries comprise 8 low-income countries (LICs), 10 lower-middle-income countries (LMICs), and two upper-middle-income countries (UMICs). GNI per capita in 2021 ranged from USD 390 (Afghanistan) to USD 7740 (Brazil). The level of GNI per capita in the most recent year conceals the economic trajectory of the countries. GNI per capita has declined over the past 10 years in 7 of the 20 countries, with precipitous declines of over 30% in 6 countries from the year GNI per capita peaked during the 2012–2021 period (Afghanistan, Angola, Brazil, Chad, Nigeria, and Mozambique) and a smaller decline of 6% in Mexico. In contrast, GNI per capita grew by more than 25% in seven countries (the DRC, Ethiopia, India, Pakistan, the Philippines, the Republic of Tanzania, and Vietnam). There have been moderate increases in GNI per capita over the past 10 years in the remaining countries (Cameroon, Indonesia, Madagascar, Myanmar, and Somalia). No data were available for the Democratic People’s Republic of Korea.

The Gross Government Health Expenditures (GGHE) per capita were considerably higher in the two UMICs (Brazil and Mexico) compared to the other countries, followed by the three LMICs from UNICEF’s East Asia and Pacific Regional Office (EAPRO) (Indonesia, Philippines, and Vietnam). The GGHE per capita was less than USD 21 in the remaining countries. External health expenditures per capita exceeded GGHE in four of the five LICs (Afghanistan, Chad, DRC, and Ethiopia) with data available, indicating the importance of development assistance for health in these countries. Out-of-pocket (OOP) health expenditures exceeded combined GGHE and external health expenditures in eight countries (Afghanistan, Cameroon, Chad, India, Myanmar, Nigeria, Pakistan, and the United Republic of Tanzania).

DPT1 coverage varied between 42 and 90%, with some of the countries having relatively high vaccination coverage, but very large birth cohorts (e.g., India and Pakistan), and other countries with relatively small birth cohorts and very low coverage (e.g., People’s Republic of Korea and Myanmar).

### 3.2. Eligibility for External Financing

The 20 countries have differential access to external financing opportunities on account of their disparate income levels and the respective prioritisation of donors. Table 2 summarises the opportunities for external financing across these 20 countries.

The lending terms available through the World Bank are determined by a country’s GNI per capita. Countries with GNI per capita below a certain threshold are eligible for support under the International Development Association (IDA). Nine of the twenty countries with the highest numbers of zero-dose children are eligible for IDA financing through the World Bank. Financing from IDA comes in the form of loans at concessional terms (0% or very low interest) and grants (which do not require repayment) to countries at a high risk of debt distress. Three of the twenty countries (Cameroon, Nigeria, and Pakistan) have per capita income below the threshold; hence, they are IDA eligible, but they are also creditworthy for lending through the International Bank for Reconstruction and Development (IBRD). These are referred to as blend countries [4]. Seven countries have access to financing through IBRD, which offers loans, guarantees, and risk management products to middle-income and creditworthy lower-middle-income countries at favourable, but not concessional, terms [5].
vaccines-11-01159-t001_Table 1Table 1Characteristics of 20 countries with the highest numbers of zero-dose children.CountryIncome GroupUNICEF RegionGNI per Capita 2012–2021GNI per Capita (2021)Gross Government Health Expenditures per Capita (2020)External Health Expenditures per Capita (2020)Out of Pocket Expenditures Per Capita (2020)DPT1 Coverage (2021)Number of Zero Dose Children (2021)AFGLICROSA
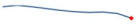
3906136074%361,000AGOLMICWCARO
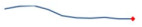
17102121957%553,000BRAUMICLACRO
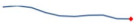
7740314115774%710,000CMRLMICWCARO
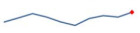
1590974076%219,000TCDLICWCARO
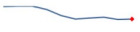
640672173%191,000PRKLICEAPRO
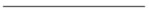




42%197,000CODLICWCARO
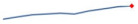
55038881%734,000ETHLICESARO
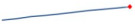
940810970%1,134,000INDLMICROSA
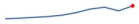
21502112988%2,711,000IDNLMICEAPRO
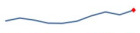
41807314274%1,150,000MDGLICESARO
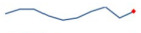
49074665%304,000MEXUMICLACRO
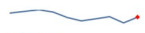
9590285020983%317,000MOZLICESARO
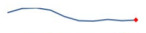
4801118367%372,000MMRLMICEAPRO
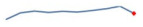
11701345645%492,000NGALMICWCARO
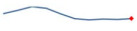
20801075270%2,247,000PAKLMICROSA
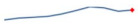
14701322190%611,000PHLLMICEAPRO
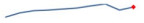
35507417457%1,048,000SOMLICESARO
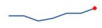
430


52%338,000TZALMICESARO
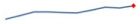
110017135682%402,000VNMLMICEAPRO
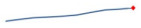
35907515287%187,000Notes: EAPRO = East Asia and Pacific Regional Office; ESARO = Eastern and Southern Africa Regional Office; LACRO = Latin America and the Caribbean Regional Office; LIC = low-income country; LMIC = lower-middle-income country; ROSA = Regional Office for South Asia; UMIC = upper-middle-income country; WCARO = West and Central Africa Regional Office. Gross government health expenditures and external health expenditures are in USD. Sources: World Development Indicators [6]; Global Health Expenditure Database [7]; WHO and UNICEF estimates of national immunisation coverage (WUENIC) [8]. Colour coding of the different types of health expenditures is based on the relative magnitude of the expenditure type (i.e. higher expenditures per capita are dark blue, while lower expenditures per capita are orange). 
vaccines-11-01159-t002_Table 2Table 2Opportunities for financing in 20 countries with the highest numbers of zero-dose children.CountryWB Lending GroupGFF EligibilityAsian Development BankGavi Co-Financing StatusGlobal Fund EligibilityIMF Poverty Reduction and Growth Trust EligibilityMomentum Routine Immunization Transformation and Equity (USAID/JSI)Bill and Melinda Gates Foundation EngagementCanGIVE & GAC-COVAX (UNICEF/Canada grant) EligibilityAFGIDAYes *YesInitial self financingYes +Yes †NoYesYesAGOIBRDYesNoFully self financingYes +NoNoNoYesBRAIBRDNoNoNever eligibleNoNoNoNoNoCMRBlendYes *NoPreparatory transitionYesYes †NoNoYesTCDIDAYesNoInitial self financingYesYes †NoYesYesPRK-NoNoInitial self financingYesNoNoNoYesCODIDAYes *NoInitial self financingYes +NoYesYesYes ‡ETHIDAYes *NoInitial self financingYes +YesYesYesYesINDIBRDYesYesFully self financingYesNoYesYesYesIDNIBRDYes *YesFully self financingYesNoNoYesYesMDGIDAYes *NoInitial self financingYes +Yes †YesNoYesMEXIBRDNoNoNever eligibleNoNoNoNoNoMOZIDAYes *NoPreparatory transitionYesYes †YesNoYes ‡MMRIDAYes *YesPreparatory transitionYesYesNoNoYesNGABlendYes *NoAccelerated transitionYes +NoYesYesYes ‡PAKBlendYesYesPreparatory transitionYesNoNoYesYesPHLIBRDYesNoNever eligibleYesNoNoNoYesSOMIDAYesNoInitial self financingYesYes †NoYesYesTZAIDAYes *NoPreparatory transitionYesYes †NoNoYes ‡VNMIBRDYes *YesFully self financingYesNoYesNoYesNotes: * Country has received support through GFF; + active direct Resilient and Sustainable Systems for Health (RSSH) grant or muti-component grant with direct RSSH component; ^†^ IMF Poverty Reduction and Growth Trust Eligibility agreement in place; ‡ Country with earmarked funds through CanGIVE. GFF = Global Financing Facility for Women, Children, and Adolescents; IBRD = International Bank for Reconstruction and Development; IDA = International Development Assistance; WB = World Bank. Color coding indicates the availability and use of particular external financing streams.

The World Bank also hosts the Global Financing Facility for Women, Children, and Adolescents (GFF), which offers catalytic grant financing (GFF Trust Fund) for leveraging domestic resources, IDA and IBRD financing, and financing from other external financing and private sector financing [9]. Seventeen of the twenty countries (all except Brazil, the Democratic People’s Republic of Korea, and Mexico) are eligible for support from the GFF, but only eleven of these have received GFF support, although some of these projects may no longer be active. Angola, Chad, India, Pakistan, the Philippines, and Somalia are eligible, but they have not yet received GFF support [10].

The Asian Development Bank (ADB) provides grants and loans to countries in Asia and the Pacific using the same operational cut-offs for concessional lending as the World Bank [11].

Gavi is the largest channel for external financing in immunisation [12]. A country’s co-financing obligations for Gavi-supported vaccines are determined by its phase of Gavi support, which is dictated by eligibility thresholds that are based on GNI per capita. Seven of the twenty countries with the highest numbers of zero-dose children are in the Initial Self-Financing Phase (Afghanistan, Chad, the Democratic People’s Republic of Korea, the Democratic Republic of Congo, Ethiopia, Madagascar, and Somalia), which has the highest subsidy level (i.e., lowest co-financing obligations). Five countries are in the Preparatory Transition Phase (Cameroon, Mozambique, Myanmar, Pakistan, and Tanzania) where co-financing obligations increase over time to reinforce country ownership and sustainability. Nigeria is in the Accelerated Transition Phase where co-financing obligations increase towards reaching full ownership upon transition. Three of the twenty countries have transitioned from Gavi vaccine support, but are still receiving project support (Angola, Indonesia, and Vietnam). Gavi has a strategic partnership with India. Brazil, Mexico, and the Philippines were never eligible for Gavi.

Except for Brazil and Mexico, all countries are eligible for support from the Global Fund to Fight AIDS, Tuberculosis, and Malaria (The Global Fund). In addition to funding disease-specific service delivery (for AIDS, Tuberculosis, and Malaria), the Global Fund also finances Resilient and Sustainable Systems for Health (RSSH) investments that strengthen system components for health in an integrated manner [13]. Six of the twenty countries have either a direct RSSH grant or a multi-component grant with an RSSH element (Afghanistan, Angola, the Democratic Republic of Congo, Ethiopia, Madagascar, and Nigeria).

Concessional lending from the International Monetary Fund (IMF) is available through the Poverty Reduction and Growth Trust (PGRT) [14]. Seven of the nine countries eligible for support through the PRGT have agreements in place (Afghanistan, Cameroon, Chad, Madagascar, Mozambique, Somalia, and Tanzania). Ethiopia and Myanmar are eligible for PGRT, but they do not have agreements.

MOMENTUM Routine Immunisation Transformation and Equity is a 5-year USAID-funded award (USD 97 million to date) to John Snow Inc (JSI) that seeks to strengthen both routine immunisation and the delivery of COVID-19 vaccines among hard-to-reach populations. Seven of the twenty countries with the highest numbers of zero-dose children are part of the MOMENTUM Routine Immunisation Transformation and Equity project.

CanGIVE is a grant to UNICEF from the Government of Canada as part of UNICEF’s ACT-A Humanitarian Action for Children (HAC) appeal. CanGIVE aims to increase COVID-19 vaccination among priority and vulnerable groups, and COVID-19 vaccine integration and recovery, including the recovery of routine immunisation services. The grant has earmarked funds for 11 countries (a total of CAD 125 million), four of which are among the 20 countries with the highest numbers of zero-dose children (the Democratic Republic of Congo, Mozambique, Nigeria, and Tanzania). In addition, CAD 85 million in unearmarked funds have been granted through GAC-COVAX to UNICEF to reach additional countries. At the time of data collection, these funds were not yet allocated.

### 3.3. External Financing Allocations, Commitments and Disbursements

Allocations, commitments, and disbursements were not available for all funding sources. All available information is summarised in Table 3, with further details on the accompanying country profiles (Appendix A).

Of the 20 countries examined, 16 countries had active health projects financed through the World Bank, excluding Brazil, the People’s Republic of Korea, Mexico, and Tanzania. Among these 16 countries, 12 had non-COVID-19-related projects and 13 countries had active projects specifically addressing the COVID-19 pandemic. Notably, in three countries (Chad, Ethiopia, and Somalia), all active World Bank health projects were COVID-19-related.

Four of the six countries eligible for ADB financing had loans or grants for PHC-related commitments (Afghanistan, India, Pakistan, and the Philippines). ADB also provided financing under its APVAX facility for COVID-19 vaccines. These commitments were not shown because the extent to which these can be reprogrammed for other purposes is unclear.

For the six countries that had a direct RSSH grant or a multi-component grant with an RSSH element, their commitments for the current implementation period are presented in Table 3. The Global Fund reports grant details for the current implementation period, which are presented in the accompanying country profiles (Appendix A). 

Other sources of funding include the CanGIVE grant to UNICEF and the Global Polio Eradication Initiative (GPEI). USD 11.7–18.0 million has been earmarked to the Democratic Republic of Congo, Mozambique, Nigeria, and Tanzania through the CanGIVE grant. Twelve of the twenty countries have access to financing from GPEI, which varied substantially across countries in 2022. Pakistan and Afghanistan had the largest allocations at USD 83.1 million and USD 36.2 million, respectively.

Table 4 shows the commitments from Gavi under HSS, CDS3, and EAF, and all funding sources overall and standardized per zero-dose child. For Gavi, potentially unallocated Gavi funds for these funding streams were calculated by subtracting approval amounts for 2021–2023 from the approval caps and ceilings [16]. For most countries, potentially unallocated Gavi funds were equal to the commitments, reflecting delays in submitting applications due to the burdens of the COVID-19 response. There is considerable variation across countries in potentially unallocated funds, which ranged from USD 46–242 per zero-dose child in 2021 in the 14 countries with HSS allocations (the countries with HSS allocations are the 13 countries currently eligible for Gavi support, and India has a strategic partnership with Gavi). There are no potentially unallocated Gavi funds in five of the seven countries that were never eligible for or have transitioned from Gavi support. The exceptions are Angola, which has a relatively small CDS3 allocation, and India, which has an HSS allocation due to its strategic partnership with Gavi. The total commitments per zero-dose child across all funding sources also varied substantially.

## 4. Discussion

We mapped active external financing for immunisation and health system strengthening, and created high-level country profiles to promote transparency and coordination to support the recovery of routine immunisation. The profiles provide a snapshot of external funding opportunities and, where possible, current active external financing.

We found that the 20 countries with the highest numbers of zero-dose children have disparate access to external financing, with the two UMICs (Brazil and Mexico) having no access to any external financing from the financing mechanism surveyed other than IBRD loans. Domestic resource mobilization is, therefore, key in these two countries, although fiscal space is likely constrained due to economic decline over the past decade, particularly in Brazil. Four additional countries (Angola, Indonesia, Philippines, and Vietnam) do not have allocations for HSS or EAF funding, but these countries are eligible for support under Gavi’s middle-income countries (MICs) approach. MICs funding can be used for activities targeting zero-dose children in Angola, Indonesia, and Vietnam, but not the Phillippines [17]. The level of potentially unallocated support from Gavi per zero-dose child varies substantially and was lowest in Nigeria and Myanmar. However, these data may under-represent the amount of Gavi funding available to reach zero-dose children. Funds that have been approved but not yet spent can be reprogrammed to reach zero-dose children. For example, in Myanmar, tens of millions of dollars of HSS funds that were dispersed are being reprogrammed to align with the zero-dose agenda.

These data provide a snapshot of the financial landscape at the time of data collection. Circumstances are rapidly changing with new commitments being made, and committed funds being approved and disbursed to countries. For example, applications to Gavi were delayed in many countries due to the burdens of the COVID-19 response. Now that the exigencies of the pandemic response have lessened, country applications are being submitted. For example, although there were no approvals for EAF funding at the time of data collection, many countries have subsequently submitted or will soon submit their applications for EAF funding.

Previous work has sought to track development assistance for health [18], maternal, newborn, and child health [19], and immunisation [12,20,21]. These tracking exercises were based primarily on the OECD Creditor Reporting System and measured past donor finance over time, or project future financing based on these data [21]. Our approach, which focuses on current donor financing, is novel.

One of the objectives of this work was to promote donor coordination to support the recovery of routine immunisation. However, a challenge in donor coordination is that different stakeholders control various financing mechanisms. For the Gavi and Global Fund, the Ministry of Health decides what is included in funding applications, whereas for the World Bank, Asian Development Bank, and IMF, the Ministry of Finance plays this role. In contrast, the bilateral (e.g., USAID) and private (e.g., Bill and Melinda Gates Foundation) donors have a more direct position in deciding how funds are used. This means that many different stakeholders would need to be engaged to ensure that funding is directed towards reaching zero-dose children.

Information on funding disbursement and utilization is crucial for greater transparency, facilitating more substantive operational discussions between governments and donors. This transparency not only enables clear and coherent planning throughout the project and grant period but also fosters accountability for both donor organizations and country governments. By ensuring that funds are used in alignment with agreed-upon priorities, transparency supports the pursuit of renewed national objectives, particularly in reaching zero-dose children. Ideally, organizations should publicly disclose comprehensive details about the funds available to countries, including the channels through which they are provided and the implementing agents responsible for administering the funds within each country. Whilst organizations, such as the World Bank and Asian Development Bank, publicly provide this information on their project documentation, other organizations provide limited information. Greater transparency in this sense helps alleviate the burden on country governments, especially in low-income countries with limited capacity for donor coordination, while also promoting greater donor–donor harmonization.

We faced key practical challenges in compiling the finance mapping data. First, limited data on active financing are publicly available. Even when data are available on public databases, it is often necessary to be in contact with respondents within organisations to gain a clear understanding of the financing mechanism and the available data. Second, where data were not routinely reported in public databases, it was often not possible to obtain data from respondents due to limitations on data being readily available within organisations’ internal grant management systems or confidentiality issues. Therefore, for some funding mechanisms, it was only possible to compile information on country eligibility. Alternatively, some organisations were able to provide the requested data, but with some delays. This affected the timeliness of reporting and may indicate that providing these data was a burden to the respondents. Third, we gathered data from the main external financing mechanisms for immunisation and PHC strengthening in these 20 countries. We may have missed some smaller funding sources that provide financing to particular countries, or where it was unclear whether the funding can be reprogrammed to strengthen PHC or immunisation. Fourth, the available data differed considerably across financing mechanisms. We followed a pragmatic approach and reported whatever data were provided to us to maximise the information conveyed in the country profiles. However, this makes it difficult to synthesise the results of the mapping across funding sources, thereby limiting transparency and hindering donor coordination.

## Figures and Tables

**Table 3 vaccines-11-01159-t003:** World Bank, Asian Development Bank, Global Fund Resilient and Sustainable Systems for Health (RSSH), Gavi, and other financings for health systems and immunisation in the 20 countries with the highest numbers of zero-dose children. Allocation and commitments as of February 2023 (USD).

	World Bank				Asian Development Bank	Global Fund	GAVI							Other	
Country	Active Health Projects	Commitments across Health Projects, Including C19	Active C19 Health Projects	Commitments across C19 Health Projects	PHC-Related Commitments	Commitments (Direct RSSH)	HSS Core Approval Cap 2021–2025	HSS Core Approved Amount 2021–2023	Equity Accelerator Fund (EAF) Core Ceiling 2021–2025	EAF Core Approved Amount 2021–2023	EAF ZIP Disbursement Forecast	CDS3 Allocation	CDS3 Approval	CanGIVE (UNICEF/Canada Grant)	GPEI
AFG	3	813,400,000	2	213,400,000	100,000,000	54,192,116 +	39,323,663	8,840,940	17,747,501	0	0	19,502,164	0	0	36,289,684
AGO	3	269,700,000	1	150,000,000	0	74,565,506 +	0	0	0	0	0	2,900,000	0	0	59,775
BRA	0	0	0	0	0	0	0	0	0	0	0	0	0	0	0
CMR	4	317,200,000	2	118,200,000	0	0	19,237,057	0	9,739,725	0	5,230,382	19,499,752	0	0	4,188,943
TCD	2	93,350,000	2	93,350,000	0	0	18,364,409	0	11,345,133	0	2,327,103	0	0	0	3,273,248
PRK	0	0	0	0	0	0	11,356,337		2,583,574	0	0	0	0	0	0
COD	7	1,201,728,134	2	247,200,000	0	39,035,627	101,554,131	40,142,271	59,730,647	0	0	0	0	13,495,746	8,378,394
ETH	3	908,600,000	3	908,600,000	0	46,545,612	99,947,139	0	44,180,347	0	6,635,585	0	0	0	1,247,636
IND	3	435,000,000	0	0	309,900,000	0	133,000,000 *	11,465,628	0	0	0	5,063,202	0	0	0
IDN	3	1,650,000,000	2	1,250,000,000	0	0	0	0	0	0	0	0	0	0	0
MDG	3	251,000,000	2	141,000,000	0	29,591,708	24,830,532	0	10,257,829	0	0	0	0	0	2,453,191
MEX	0	0	0	0	0	0	0	0	0	0	0	0	0	0	0
MOZ	4	1,235,160,000	2	215,000,000	0	0	31,250,799	13,903,670	10,515,600	0	0	12,282,775	0	13,505,367	16,759,228
MMR	2	200,000,000	0	0	0	0	17,935,978	0	5,529,571	0	0	0	0	0	0
NGA	3	1,550,000,000	2	900,000,000	0	148,320,456	126,000,000	39,992,387	0	0	6,542,899	27,757,554	0	18,038,654	13,974,830
PAK	7	736,300,000	1	200,000,000	100,000,000	0	114,957,670	650,000	45,686,028	0	0	0	0	0	83,125,000
PHL	3	1,700,000,000	3	1,700,000,000	125,000,000	0	0	0	0	0	0	0	0	0	0
SOM	2	91,000,000	2	91,000,000	0	0	26,406,593	4,450,330	13,334,053	0	7,152,834	0	0	0	7,484,388
TZA	0	0	0	0	0	0	31,615,684	0	15,707,813	0	0	0	0	11,667,132	19,425,759
VNM	1	88,000,000	0	0	0	0	0	0	0	0	0	0	0	0	0

Notes: + Multicomponent grant with a direct RSSH component. * India has a special Board Approved strategy and has an HSS allocation of USD 133 million for 2022–2026 [15]. The HSS Core Approval Cap is the maximum amount a country can access in approvals for disbursements during Gavi 5.0 (2021–2025) for the health system and immunisation strengthening support. The Equity Accelerator Fund Ceiling is the maximum amount the country can access for approvals during Gavi 5.0 (2021–2025) [16]. EAF ZIP disbursement forecast is an estimate. CDS3 is the third window of support under COVID-19 Vaccine Delivery Support (CDS). The third objective of CDS3 is to support the integration of COVID-19 vaccine with routine immunisation to achieve sustainable benefits. It should be noted that, for Gavi financing, countries receive information on grant ceilings to budget against for their specific grant period, which may not align with the years covered by the Gavi 5.0 strategic period and, as such, refer to different amounts than stated in the table. Information related to HSS, EAF, and CDS3 financing is shown in Table 3, with information on other Gavi funding streams shown in the country profiles (Appendix A). EAF financing comprises two components: USD 400 million for core allocations and USD 100 million to consortiums of Civil Society Organisations and expanded partners to deliver vaccines in fragile settings as part of the Zero-dose Immunisation Programme (ZIP) in the Sahel and Horn of Africa. Country allocations for Core EAF funding, along with estimates of the disbursement forecast by the country for EAF ZIP funding are shown (ZIP Disbursement forecasts by country were available for 18 months after implementation from January 2023 to June 2024. An estimate for the entire implementation period (January 2023–December 2025) was calculated by assuming that funds allocated beyond June 2024 would be disbursed to countries in the same proportion as in the initial forecast). At the time of data collection, no HSS funds were approved during 2021–2023 in 7 of the 14 countries with HSS allocations (Cameroon, Chad, People’s Republic of Korea, Ethiopia, Madagascar, Myanmar, and Tanzania). There were also no approvals for EAF Core or CDS3 funds in any of the countries at the time of data collection.

**Table 4 vaccines-11-01159-t004:** Total commitments from Gavi and all funding sources overall and per zero-dose child as of February 2023.

	GAVI				All Funding Sources	
Country	Total HSS + EAF + CDS3 Commitments	HSS + EAF + CDS3 Commitments per Zero Dose Child	Potentially Unallocated HSS + EAF + CDS3 Funds	Potentially Unallocated Funds per Zero-Dose Child	Total Commitments	Total Commitments per Zero Dose Child
AFG	66,822,246	185	57,981,306	161	1,152,436,560	3192
AGO	1,450,000	3	1,450,000	3	347,225,281	628
BRA	-	0	-	-	-	0
CMR	43,957,040	201	43,957,040	201	384,513,122	1756
TCD	32,036,645	168	32,036,645	168	148,035,470	775
PRK	13,939,911	71	13,939,911	71	17,359,435	88
COD	161,284,778	220	121,142,507	165	1,473,362,755	2007
ETH	150,763,071	133	150,763,071	133	1,150,230,348	1014
IND	135,531,601	50	124,065,973	46	2,882,963,202	1063
IDN	-	0	-	-	2,100,000,000	1826
MDG	35,088,361	115	35,088,361	115	328,898,883	1082
MEX	-	0	-	-	-	0
MOZ	47,907,787	129	34,004,117	91	1,331,438,140	3579
MMR	23,465,549	48	23,465,549	48	231,587,228	471
NGA	146,421,676	65	106,429,289	47	1,916,843,069	853
PAK	160,643,698	263	159,993,698	262	1,623,781,687	2658
PHL	-	0	-	-	3,025,000,000	2886
SOM	46,893,480	139	42,443,150	126	164,157,971	486
TZA	47,323,497	118	47,323,497	118	91,194,668	227
VNM	-	0	-	-	88,000,000	471

Notes: Potentially unallocated Gavi funds through HSS, EAF, and 50% of CDS3 funds per country were calculated by subtracting approval amounts for 2021–2023 from the allocations (i.e., approval caps and ceilings). This amount may include funding that is already planned for, but not yet allocated (e.g., where country proposals have been submitted, but not yet approved). In some cases, planning for funding and even approval and disbursement funding may still be reprogrammed by countries to different priorities and, hence, could be repurposed for reaching zero-dose children. All financing amounts are in USD. Colour coding of funds per capita is based on their relative magnitude (i.e., higher commitments per capita are dark blue, while lower commitments per capita are dark orange).

## Data Availability

All data presented in the paper are available in the Appendix A.

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
