# Peer review of "Recovery of Routine Immunisation: Mapping External Financing Opportunities for Reaching Zero-Dose Children"

_vaccines, 2023, doi:10.3390/vaccines11071159_

Round 1

Reviewer 1 Report

The authors present the problem of strengthening health care and mapping external funding sources for immunization.This is a well-known problem, although the authors presented it in an interesting way.
The authors mapped active external sources of funding for immunization and health care strengthening, and profiled countries to promote transparency and coordination to support the restoration of routine immunization. It is interesting to note that among the 20 countries with the highest number of zero-dose children, many have varying access to external funding, and two UMICs (Brazil and Mexico) have no access to any external funding beyond IBRD loans.
Unfortunately, the results of the paper show that data may be incomplete and available in a limited way, as the authors point out in the discussion. This affects the results presented, but is typical of this type of study.
Hence, the conclusion of the paper, which is the last sentence in the discussion, should be emphasized more strongly, not only in the discussion but also in the abstract. The authors should indicate, the possibility of transferring the assumptions of their work to other undervaccination of children in relation to other common childhood diseases.

Author Response

We have added this point to the abstract. We have also expanded the discussion to provide a more comprehensive overview of the importance of financing data for transparency and donor coordination, and the implications of having limited publicly available data. In addition, we have extended the analysis of commitments and allocations from Gavi and all funding sources per zero-dose child in Table 4.

A zero-dose child is a child that has never received any vaccine. To facilitate identification of children that lack access to routine immunization, the internationally accepted operational definition of a zero-dose child is a child that has not received a single dose of DPT-containing vaccines. Therefore, the work is relevant to the undervaccination of children generally.

Reviewer 2 Report

I think that it is an interesting and focuses on an important problem of developing countries in children vaccination covering especially after pandemic. In my opinion there are no obvious flags or weakness area. Conclusions are coherent with bulk of paper.

Author Response

We thank the reviewer for the comments. 

Reviewer 3 Report

In the manuscript by Tougher et al, the authors present (and analyze) the eligibility of 20 countries with known childhood vaccination deficiencies (i.e. Zero-dose Children).  By comparing the eligibility of these countries to various worldwide vaccination programs (i.e. donor financing funded by private and international bodies) the authors attempt to provide an overview of the potential resources made available to these 20 countries, with the objective of providing the reader an idea of the ability of the aforementioned countries to combat challenges in childhood vaccination (using DTP vaccination as a primary measurement for the classification of Zero-dose countries).

While the topic is interesting, the Reviewer feels that the qualitative approach used by the authors are not suitable for Vaccines.  That is not to say that this paper is poor; it is simply that the data presented is too generalized (and somewhat vague) to merit publication in a scientific journal.  The Reviewer feels that this manuscript would be best suited for a journal more specialized in Economics/Social Sciences (in fact, there might even be some Public Health journals that would also find this subject to be quite relevant).

It is the hope of the Reviewer that this manuscript be published; however, it would be more appropriate if it were published in another journal (with a different scope).  The Reviewer commends the authors for the work for UNICEF, and wish them the best of luck in their submission to another journal. 

Author Response

We thank the reviewer for their comments.

The paper is quantitative and descriptive. The analysis of donor financing for immunization and is an active area of research in the field of public health. For example, recent studies of this type include:

  • Sriudomporn S, Sim SY, Mak J, Brenzel L, Patenaude BN. Financing And Funding Gap For 16 Vaccines Across 94 Low- And Middle-Income Countries, 2011-30. Health Aff (Millwood). 2023 Jan 1;42(1):94–104.

  • Ikilezi G, Micah AE, Bachmeier SD, Cogswell IE, Maddison ER, Stutzman HN, et al. Estimating total spending by source of funding on routine and supplementary immunisation activities in low-income and middle-income countries, 2000-17: A financial modelling study. The Lancet. 2021; 398:1875–93. Available from: https://doi.org/10.1016/

  • Pitt C, Grollman C, Martinez-Alvarez M, Arregoces L, Borghi J. Tracking aid for global health goals: a systematic comparison of four approaches applied to reproductive, maternal, newborn, and child health. Lancet Glob Health. 2018 Aug 1;6(8):e859–74. Available from: http://www.thelancet.com/article/S2214109X18302766/fulltext

  • Ikilezi G, Zlavog B, Augusto OJ, Sherr K, Lim SS, Dieleman JL. Tracking donor funding towards achieving the Global Vaccine Action Plan (GVAP) goals: A landscape analysis (1990-2016). Vaccine. 2018 Nov 26;36(49):7487–95. Available from: https://pubmed.ncbi.nlm.nih.gov/30366804/

These existing studies were based primarily on the OECD Creditor Reporting System and measure past donor finance over time, or project future financing based on the OECD data. Our methods, which focus on current donor financing, are novel and more relevant to for donor coordination and transparency in response to recovery of a health emergency. Consequently, we feel that this paper is well-suited for publication in Vaccines and fits well within the purview of the Immunization Strategies and Vaccine Uptake after the SARS-CoV-2 Pandemic Special Issue.

Round 2

Reviewer 3 Report

While the paper has some merit for publication (i.e. in another journal), the Reviewer still doesn't believe that it would be suitable for vaccines.  However, this is an issue that would ultimately be decided by the editors.